# Phenotypic Screen with the Human Secretome Identifies FGF16 as Inducing Proliferation of iPSC-Derived Cardiac Progenitor Cells

**DOI:** 10.3390/ijms20236037

**Published:** 2019-11-30

**Authors:** Karin Jennbacken, Fredrik Wågberg, Ulla Karlsson, Jerry Eriksson, Lisa Magnusson, Marjorie Chimienti, Piero Ricchiuto, Jenny Bernström, Mei Ding, Douglas Ross-Thriepland, Yafeng Xue, Diluka Peiris, Teodor Aastrup, Hanna Tegel, Sophia Hober, Åsa Sivertsson, Mathias Uhlén, Per-Erik Strömstedt, Rick Davies, Lovisa Holmberg Schiavone

**Affiliations:** 1Bioscience Cardiovascular, Research and Early Development, Cardiovascular, Renal and Metabolism, BioPharmaceutical R&D, AstraZeneca, Gothenburg, 43150 Mölndal, Sweden; karin.jennbacken@astrazeneca.com (K.J.); Jerry.eriksson@astrazeneca.com (J.E.); lisa.magnusson@astrazeneca.com (L.M.); marjorie.chimienti@orange.fr (M.C.); 2Mechanistic Biology and Profiling, Discovery Sciences, R&D, AstraZeneca, Gothenburg, 43150 Mölndal, Sweden; fredrik.wagberg@astrazeneca.com (F.W.); ulla15405@gmail.com (U.K.); Per-Erik.Stromstedt@astrazeneca.com (P.-E.S.); 3Quantitative Biology, Discovery Sciences, R&D, AstraZeneca, Cambridge CB40WG, UK; piero.ricchiuto@astrazeneca.com; 4Discovery Biology, Discovery Sciences, R&D, AstraZeneca, Gothenburg, 43150 Mölndal, Sweden; jennybernstrom@hotmail.com (J.B.); Mei.Ding@astrazeneca.com (M.D.); 5Discovery Biology, Discovery Sciences, R&D, AstraZeneca, Cambridge CB40WG, UK; Douglas.Ross-Thriepland@astrazeneca.com (D.R.-T.); Rick.Davies@astrazeneca.com (R.D.); 6Structure and Biophysics and Fragment screening, Discovery Sciences, R&D, AstraZeneca, Gothenburg, 43150 Mölndal, Sweden; yafeng.xue@astrazeneca.com; 7Attana AB, 11419 Stockholm, Sweden; diluka.peiris@attana.com (D.P.); teodor.aastrup@attana.com (T.A.); 8Department of Protein Science, School of engineering Sciences in Chemistry, Biotechnology and Health, KTH-Royal Institute of Technology, 11421 Stockholm, Sweden; hannat@kth.se (H.T.); sophia@kth.se (S.H.); asa.sivertsson@scilifelab.se (Å.S.); mathias.uhlen@scilifelab.se (M.U.)

**Keywords:** human secretome, phenotypic screening, cardiac progenitor cells, fibroblast growth factors, fibroblast growth factor 16

## Abstract

Paracrine factors can induce cardiac regeneration and repair post myocardial infarction by stimulating proliferation of cardiac cells and inducing the anti-fibrotic, antiapoptotic, and immunomodulatory effects of angiogenesis. Here, we screened a human secretome library, consisting of 923 growth factors, cytokines, and proteins with unknown function, in a phenotypic screen with human cardiac progenitor cells. The primary readout in the screen was proliferation measured by nuclear count. From this screen, we identified FGF1, FGF4, FGF9, FGF16, FGF18, and seven additional proteins that induce proliferation of cardiac progenitor cells. FGF9 and FGF16 belong to the same FGF subfamily, share high sequence identity, and are described to have similar receptor preferences. Interestingly, FGF16 was shown to be specific for proliferation of cardiac progenitor cells, whereas FGF9 also proliferated human cardiac fibroblasts. Biosensor analysis of receptor preferences and quantification of receptor abundances suggested that FGF16 and FGF9 bind to different FGF receptors on the cardiac progenitor cells and cardiac fibroblasts. FGF16 also proliferated naïve cardiac progenitor cells isolated from mouse heart and human cardiomyocytes derived from induced pluripotent cells. Taken together, the data suggest that FGF16 could be a suitable paracrine factor to induce cardiac regeneration and repair.

## 1. Introduction

Heart failure is a major cause of mortality worldwide with a 46% projected increase in prevalence by 2030 [1]. Myocardial infarction (MI) is one common cause of heart failure and results in the loss of a substantial number of cardiomyocytes, which are replaced by fibrotic scar tissue. The available therapies for these patients are symptomatic and can only slow down disease progression and reverse limited aspects of cardiac dysfunction. Lower vertebrate species such as zebrafish and newts can regenerate new cardiac tissue after an injury, which was also shown in rodent neonatal heart [2,3,4]. Since cardiomyocytes in the adult mammalian heart have limited regenerative capacity (< 1%), different strategies are being investigated to promote cardiac repair. This includes cell therapy using pluripotent stem cells, administration of exogenous paracrine factors in the form of recombinant proteins or modified messenger RNA, and other modalities to stimulate cardiomyocyte proliferation, progenitor cell proliferation, and angiogenesis (for a review, see References [5,6,7,8,9]).

The secretome was shown to be a modulator of the regenerative response and can induce repair via expansion of various cardiac populations including cardiac progenitor cells or endothelial cells, as well as having anti-apoptotic, anti-fibrotic, and immunomodulatory effects. Cardiac progenitor cells (CPCs) present in the heart could potentially contribute to regeneration via secretion of reparative paracrine factors or by differentiation into endothelial cells and vascular cells; however, these cells have limited or no cardiomyogenic potential [10,11,12,13,14]. The secretome from different cell sources was shown to induce cardiac repair and improvement in cardiac function after intramyocardial injection in mouse post MI [15]. In addition, conditioned medium from mesenchymal stem cells was shown to reduce infarct size in a porcine model of ischemia–reperfusion injury [15,16] which highlights that paracrine factors are of importance for cardiac repair.

To be able to identify secreted paracrine factors that affect a specific disease-relevant process, we applied an omics approach to generate the human secretome in mammalian cell factories [17] that can then be used in phenotypic screening for identification of new targets. Here, we used a library of 923 secreted proteins to identify factors that stimulate proliferation of human-induced pluripotent cardiac progenitor cells (iPS-CPC) (enriched population of KDRpositive (vascular endothelial growth factor [VEGF] receptor 2)/PDGFRαpositive (platelet-derived growth factor receptor alpha)/NKX2.5positive/CKITnegative CPCs). We identified several fibroblast growth factor (FGF) proteins that stimulated proliferation of human cardiac progenitor cells, and two of the identified proteins belong to the FGF9 subfamily, which is suggested to be involved in cardiac repair [18]. Focused follow-up work on FGF16 showed that the proliferative effect of FGF16 specifically induces proliferation of cardiac progenitor cells and cardiomyocytes and suggests that FGF16 has different receptor preferences on cardiac cells compared to FGF9. This work also demonstrates the feasibility of using the secretome in phenotypic screening to identify key factors involved in regenerative processes.

## 2. Results

### 2.1. FGF9, FGF16, and 10 Additional Secreted Proteins Identified to Stimulate CPC Proliferation in a Phenotypic Screen

To enable identification of pathways and targets involved in proliferation of CPCs, a previously established medium-throughput 384-well assay using iPS-derived hCPCs [19] (Figure 1A) was used for screening a human secretome library. The library (Appendix A), consisting of 923 unique secreted proteins, was added to hCPCs at three different concentrations, and cell proliferation was measured after three days. The data from the primary screen are summarized in Figure 1B.

Active proteins, based on an activity threshold of 30% stimulation of proliferation above baseline control, were subsequently tested in a 10-point concentration response (CR) in triplicate. This led to confirmation of 12 unique active proteins with varying potencies. Most active proteins belonged to the FGF family. FGF9 was identified to have the strongest maximal effect on hCPC proliferation (> 100% compared to the positive control and half maximal effective concentration, EC_50_, around 1 nM) (Figure 2A, black circles). The closely related FGF9 family member FGF16 was also identified as active but with an EC_50_ of 10 nM (Figure 2B). Additional FGF proteins that were identified as active were FGF1, FGF4, and FGF18. In addition, Noggin (Figure 2C), NPTX1 (Neuronal Pentraxin 1; Figure 2D), VEGF-A (Vascular endothelial growth factor A; Figure 2E), and PDGF-C (Platelet-derived growth factor-C), PDGF-A, ECM1 (Extracellular matrix protein 1), and VMO1 Vitelline membrane outer layer protein 1 homolog) secretome proteins induced hCPC proliferation.

### 2.2. FGF16, Noggin, NPTX1, and VEGF-A Specifically Induce Proliferation of Human Cardiac Progenitor Cells but not Human Cardiac Fibroblasts

To understand if the effect of the identified active proteins was specific for hCPCs, a counter-screen with human primary cardiac fibroblasts (hCFs) was also performed on all confirmed active proteins. In this assay, secretome proteins were incubated with human cardiac fibroblasts, and proliferation was measured by nuclear count after eight days as previously described [20]. The data showed that FGF16 (Figure 2B), Noggin (Figure 2C), NPTX1 (Figure 2D), and VEGF-A (Figure 2E) were specific for proliferation of hCPCs, whereas some of the active proteins, including FGF9 (Figure 2A, white circles), also induced proliferation of hCFs.

### 2.3. Biosensor Analysis Suggests That FGF9 and FGF16 Bind to Different Receptors on hCPCs and hCFs

Surprisingly, FGF9 and FGF16, which belong to the FGF9 subfamily and have high sequence identity (Appendix A) and similar preferences for FGF receptors [21], induced different proliferative effects on hCFs. To elucidate receptor preferences, we used a quartz crystal microbalance (QCM) biosensor to study the binding of FGF9 and FGF16 to hCPCs and hCFs. This allowed us to follow the interaction in real time and to calculate binding parameters such as K_a_, K_d_, K_D_, and B_max_ [22]. hCPCs and hCFs were grown on the QCM COP-1 surface. FGF9 and FGF16, at several concentrations, were passed over the cell surface, and the responses were monitored. For both FGF9 and FGF16, the data suggested a 1:2 interaction to hCPCs (Figure 3A,B, left traces, and Appendix A). FGF9 displayed similar types of interactions with hCFs (Figure 3A, right trace, and Appendix A).

In contrast, the FGF16 binding curve for hCFs showed different characteristics with only a weak 1:1 interaction detected for the hCFs (Figure 3B, right trace). Repeated injections of either FGF9 or FGF16, close to saturation, did not give an additive effect on either cell type (Figure 4A). However, addition of a mixture of FGF16 and FGF9, after saturation with either ligand, gave an additional response which was similar to the response elicited by each of the proteins injected individually (Figure 4B,C and Appendix A). FGF9 displayed a similar interaction with both hCPCs and hCFs, while FGF16 had a much stronger interaction with hCPCs than hCFs (Appendix A). This suggested that FGF9 and FGF16 bind to different FGF receptors on the hCPCs and hCFs. The calculated receptor density, for the interacting receptors, based on the number of cells on the COP-1 surface (Appendix A), supports the B_max_ values, with a more than 10-fold higher receptor density for the FGF16 interaction with hCPCs than with hCFs (Appendix A).

To better understand to what extent the FGF receptors are expressed on hCPCs and hCFs, we performed quantitative PCR analysis of FGF receptor expression. This showed that *FGFR1* (fibroblast growth factor receptor 1) is expressed on both iPSC-derived hCPCs and hCFs (Figure 5). *FGFR2* is specifically expressed on hCPCs, although at lower amounts, and there is a 10-fold lower expression of *FGFR4* and *FGFR3* on hCPCs and no detectable expression on hCFs.

### 2.4. FGF16 Induces Proliferation of Naïve CPCs Isolated from Mouse Hearts and Human iPS-Derived Cardiomyocytes

The primary screen was performed on human CPCs derived from iPS cells, and, to investigate if the effect was translatable to native CPCs, we also tested the effects of FGF16 on CPCs isolated from mouse heart. Both human and mouse FGF16 recombinant proteins induced proliferation of Sca1- positive mouse CPCs, measured as an increase in nuclear count (Figure 6A), confirming translation of the findings. Cardiomyocytes emerged as a main contributor to formation of new cardiac tissue, and FGF16 recombinant protein was, therefore, also tested in a proliferation assay with human iPS-derived cardiomyocytes. The data show (Figure 6B) that, in addition to inducing CPC proliferation, FGF16 stimulated cell-cycle activity in human cardiomyocytes as measured by EdU (5-Ethynyl-2´-deoxyuridine) incorporation.

## 3. Discussion

Different approaches, which include cell therapy, stimulation of endogenous cardiac progenitor cells, or stimulation of cardiomyocyte proliferation, are being explored to regenerate new cardiac tissue after myocardial damage. Cardiac progenitor cells are rare in the adult heart; however, previous studies showed that they could contribute to repair of the myocardium via differentiation to vascular cells and via secretion of paracrine factors, which altogether have favorable effects on the damaged myocardium [10,11,12,13,14]. In the absence of adult human cells for testing, human iPS-derived cells can be used as a model system to investigate and discover novel biology. Here, we performed a phenotypic screen with a human secretome library and identified promising hits that stimulated proliferation of human cardiac progenitor cells. The results show that a large portion of the active proteins in the screen belong to the FGF family of proteins.

We show that FGF16 specifically induced proliferation of CPCs and induced cell-cycle activity in human cardiomyocytes. The data highlight a novel effect on CPCs and show that FGF16 overall is an interesting factor to explore for cardiac regeneration and repair, since it also stimulates cardiomyocytes. In contrast, FGF9 had a more general proliferative effect on cardiac progenitor cells, cardiac fibroblasts, and kidney fibroblasts (Figure 2). We previously used human iPS-derived CPCs in phenotypic high-content screening assays in our team to identify small-molecule compounds that induce proliferation [19,23]. In the presented work, we focused on screening a human secretome library in a proliferation assay with hCPCs. It is unlikely that the hCPC differentiated into other cell types during this time frame, and Nkx2.5 stainings were performed on selected proteins, also confirming that a majority of cells at three days post plating were at the cardiac progenitor cell stage. However, we cannot exclude the possibility that secretome proteins induced differentiation into other cell types. CPC differentiation into cardiomyocytes, smooth muscle cells, endothelial cells, or fibroblasts can be captured, but requires an extended incubation for at least seven days [19,24].

Expression analysis suggests that FGF16 is present in the adult heart [25,26,27], and previous data also showed that FGF16 is involved in other mechanisms important for regeneration. FGF16 knock-out mice are normal and fertile, but heart weight and cardiomyocyte numbers are reduced at six months of age indicating an important role of FGF16 in controlling cardiomyocyte numbers [28]. At the same time, cardiac-specific overexpression of FGF16 in neonatal mouse heart subjected to cryoinjury induced cardiomyocyte replication and improved heart function in vivo [29]. Intramyocardial injection of FGF16 recombinant protein in db/db mice after myocardial infarction improved cardiac function and reduced interstitial fibrosis and monocyte infiltration after two and four weeks. FGF16 could also have a protective role in an acute setting of cardiac injury since doxorubicin treatment in mice with FGF16 knockdown decreases heart systolic function and increases cell death compared to wild-type controls. In contrast, overexpression of FGF16 or treatment with a single bolus of recombinant FGF16 increased left-ventricular contractility caused by acute doxorubicin treatment in a mouse model [30]. In support for the role of FGF16 in rodents, an increased incidence of cardiovascular abnormalities was observed in a family carrying a nonsense mutation in the *FGF16* gene [31].

The biosensor data presented here suggest that FGF9 and FGF16 could signal through different receptors and provide an explanation for the different proliferative effects observed on cardiac fibroblasts. Messenger RNA (mRNA) expression profiling in mouse heart suggests that there is high expression of FGF16 and FGFR1c and moderate expression of FGF9 and FGFR2c [27]. We detect high mRNA expression levels of FGFR1 on both hCPCs and hCFs, whereas FGFR2 is only detected on the hCPCs and FGFR3 and FGFR4 expression levels are low. Based on the receptor mRNA levels detected, we speculate that FGF9 binds to FGFR1c, whereas FGF16 binds to FGFR2c. Sequence analysis (Appendix A) shows that, while the core region of FGF9 and FGF16 is highly conserved, the N-terminal 51 amino acids have low sequence identity. This could provide a basis for a preference for different receptor interactions on a cell expressing several different FGF receptors simultaneously (detailed in Appendix A).

Another interesting candidate that was identified in the hCPC screen was VEGF-A. Previous studies showed that VEGF-A plays a critical role in the regulation and formation of new blood vessels and has a pro-survival effect on vascular and endothelial cells [32]. VEGF-A also expands mouse epicardial-derived progenitor cells and induces differentiation toward endothelial cells, thereby playing a critical role in regeneration [33]. We previously showed that VEGF-A administered as a single injection of modified mRNA improves systolic ventricular function and limits myocardial damage in small and large animals in permanent occlusion myocardial infarction models. Both arteriolar and capillary density increased at two months post treatment, and interstitial fibrosis was decreased in the pig model [6]. Our findings that VEGF-A also induces proliferation of human CPCs strengthen its role in cardiac regeneration and show that it is an important factor not only for endothelial cells and epicardial-derived cells but also for cardiac progenitor cells.

Two antagonists to TGFβ (transforming growth factor β) superfamily signaling were identified as active proteins in the hCPC screen. Noggin is a well-described BMP (bone morphogenetic protein) signaling antagonist [34] of Activin/Nodal and BMP4 [35] and is an important regulator during heart development. Furthermore, it directly and indirectly controls proliferation and differentiation of cardiac cells. NPTX1 is also reported to be a BMP antagonist, which is involved in neural induction of human pluripotent stem cells by reducing both Nodal and BMP signaling [36]. To our knowledge, the proliferative effect of NPTX1 on cardiac progenitor cells reported here is novel.

## 4. Materials and Methods

### 4.1. Selection of Protein Genes for Inclusion in Secretome Library

Secreted proteins were defined as all Uniprot entries having subcellular location “Secreted”, in addition to all genes with at least one transcript predicted to be secreted according to the Human Protein Atlas (HPA). For prediction of secreted proteins in HPA, three different signal peptide prediction algorithms [37,38,39], in combination with seven different transmembrane (TM) region prediction algorithms [40], were used. To be categorized as secreted, a transcript must have a signal peptide predicted by at least two of three methods, and no TM region predicted by four or more methods. Selected extracellular domains (ECDs) were also included in the secretome library. One-pass TM proteins for production of ECDs were selected from Uniprot entries with subcellular location “one-pass TM proteins”, as well as from HPA TM region predictions [40]. Additional general stratification was made to order the making of the library using different databases (GeneOntology: “Extracellular space” annotation, Ingenuity Pathway Analysis (IPA): “Extracellular space” annotation, Uniprot: proteins with a SignalP (and no T region)) or proteins annotated as “Secreted” and also based on “The Human Secretome Atlas” [41]. Specific stratifications were made for cardiac cells [42,43,44,45,46].

### 4.2. Protein Production and Quality Control

Secretome proteins were expressed in Chinese Hamster Ovary (CHO) EBNALT 85 cells using the QMCF Technology (Icosagen Cell Factory OÜ, Tartu, Estonia). The expression cassette was based on the CMV promoter, an N-terminal CD33 signal peptide for secretion of all produced proteins and a C-terminal HPC4 (human protein C4) tag for affinity purification using an anti-HPC4 coupled resin and mild elution with EDTA [17]. After desalting, the concentration of the proteins was determined. The average concentration of the produced proteins after desalting was 13.5 µM. The proteins were then aliquoted, and all proteins with a concentration > 9 µM were diluted to 9 µM with 1× PBS before they were snap-frozen in liquid nitrogen and stored at −80 °C for a short period of time until the material was sub-aliquoted for long-term storage. Proteins included in the tested library were at measured concentrations that varied between 2 µM and 9 µM. During the aliquotation, a sample was collected for quality control. Each purified protein was identified by MS/MS. The purity of the proteins was analyzed using SDS-PAGE and Western blot using an anti-protein C antibody. Glycosylation patterns of the purified proteins were also analyzed using SDS-PAGE. Endotoxin levels were generally < 0.5 EU (endotoxin units) /mL. Some proteins failed during different stages of the production process. For example, 13 of the 18 secreted FGFs (FGF1, FGF4, FGF5, FGF6, FGF7, FGF9, FGF10, FGF16, FGF17, FGF18, FGF19, FGF21, and FGF23) were part of the tested secretome library, whereas the remaining FGF2, FGF3, FGF8, FGF20, and FGF22 could not be included in the screening due to failure during protein production. Human and mouse FGF9 and FGF16 were scaled up using a CHO–EBNA–GS system [47] using the pEBNAZ vector [48] and a C-terminal His-tag for purification with Ni-NTA chromatography followed by size-exclusion chromatography.

### 4.3. Sample Management of Proteins

For final aliquoting, each protein batch was thawed once and dispensed into 15–20-µL aliquots in 0.5-mL FluidX tubes (66-52325-Y6; Brooks Automation, Inc. MA, US) before snap-freezing and long-term storage. Protein batches were not refrozen after the second freeze–thaw. Before testing in screens, secretome proteins were dispensed and diluted in PBS in 384-well V-bottom plates before addition to cell-based assays. A Microlab STAR automated liquid handling workstation (Hamilton Company U.S.) was used for the creation of protein aliquots and 384-well plates for cell assays. Furthermore, 384-well Grenier V-bottom plates (784201) from Greiner Bio One International GmbH (Germany) were used for dispensing and dilution of secretome proteins. The average concentration of produced proteins was 13.5 µM.

However, all proteins with a concentration above 9 µM were diluted to 9 µM. Based on these numbers the average top concentration of protein in the cell assays was ~400 nM. The proteins were tested in varying dilution series (e.g., in the hCPC assay, three concentrations were tested as 10-fold dilutions, with final assay concentrations typically of 400, 40, and 4 nM).

All information about the individual proteins in the library (i.e., gene name, sequence, concentration, and quality-check report) was maintained in a laboratory information system at the KTH Royal Institute of Technology, from where it was exported to AstraZeneca’s implementation of the Labguru application (BioData, http://www.labguru.com//) that was used to share information about pre-clinical bioreagents. The library of proteins was further registered in AstraZeneca’s compound management databases (internal AstraZeneca software and Mosaic, https://www.titian.co.uk), originally used for small molecules but now expanded to handle proteins, to allow for seamless integration between compound handling, assay screening, and data analysis. Cross-referencing of the databases allowed full traceability of results and information.

### 4.4. Human iPSC-Derived CPC Culture and Proliferation Screen

Cryopreserved human iPSC-derived CPCs (CPC-301-020-001-PT) were from Cellular Dynamics International (CDI, Madison, WI, USA) and stored at −150 °C until use [19]. Briefly, cryopreserved human iPSC-CPCs were thawed and transferred to hCPC assay medium (William’s E medium, A1217601, supplemented with Cell Maintenance Cocktail B, CM4000), according to the manufacturer’s instructions (ThermoFisher Waltham, MA, USA). Cells were seeded in fibronectin-coated 384-well Corning plates (3712/3764) at a density of 6000 cells/well in 45 µL of hCPC assay medium. The plates were incubated for 24 h at 37 °C. On the day of dosing secretome proteins, 25 µL of the medium was removed and 2 µL of secretome proteins (stock solutions, PBS diluted 14190-094 ThermoFisher) or 2 µL of 2.5 µg/mL FGF2 positive control (PHG026 ThermoFisher) or negative control PBS were added to individual wells, followed by addition of 25 μL of fresh hCPC medium. After 72 h of treatment, cells were fixed for 20 min at room temperature by adding 25 µL of 11.2% formaldehyde in PBS to each well without removing any medium and washed with PBS; then, 25 µL of blocking buffer containing 1% BSA, 0.1% Triton X-100, and Hoechst 33342 (ThermoFisher) was added (1:2000) for 30 min at RT, followed by final washing with PBS. The plates were sealed and imaged on an ImageExpress Micro (Molecular Devices, Berkshire, UK) using 10× objective. Images were analyzed for nucleus counts using a MetaXpress software (Molecular Devices).

Antibody staining with Nkx2.5 was performed as a control for a few selected proteins that induced proliferation to check that the cells did not differentiate into any other cell type. Cells were treated, fixed, and washed as described above and stained with Nkx2.5 antibody (dilution 1/200, ab91196 Abcam, Cambridge UK) overnight at 4 °C, before being washed and incubated for 45 min at RT with Alexa Fluor 488-conjugated secondary antibody (ThermoFisher) and Hoechst 33342 (1:2000), followed by final washing with PBS. The plates were sealed and imaged on an ImageExpress Micro (Molecular Devices) using 10× objective.

### 4.5. Human Native Cardiac Fibroblasts Culture and Proliferation Assay

Cryopreserved human primary hCF cells (C12375) were from PromoCell (Heidelberg, Germany). Before screening, human hCF cells were thawed, expanded in complete fibroblast growth medium (fibroblast growth medium including supplement, C23130, PromoCell), and frozen at 1 × 10^6^ cells/vial and stored at −150 °C until further use. Cryopreserved expanded hCFs were thawed and pre-cultured for six days in complete fibroblast growth medium, and used in the screen as described previously [20]. Briefly, pre-cultured cells were harvested using TrypLE Express (12604, ThermoFisher) and seeded in 384-well Falcon plates (353962, Corning) at a density of 750 cells/well in 45 µL of assay medium. The plates were left for 30 min at RT to allow cells to settle, followed by 24 h of incubation at 37 °C. On the day of dosing secretome proteins, 25 µL of the medium was removed, and 2 µL of secretome proteins (stock solutions or PBS diluted) or negative control PBS were added to individual wells, followed by addition of 25 µL of fresh assay medium. After 72 h, 40 µL of assay medium was added to each well. The plates were cultured for another five days, after which 45 µL of medium was removed and cells were fixed for 30 min at RT by adding 25 µL of 11.2% formaldehyde to each well, washed with PBS, permeabilized with 0.1% Triton X-100 in PBS, and stained with Hoechst 33342 (1:2000). The plates were sealed and imaged on an ImageExpress Micro (Molecular Devices) using 4× objective. Images were analyzed for nucleus counts using the MetaXpress software (Molecular Devices).

### 4.6. Isolation of Mouse Sca1-Positive Progenitor Cells from Naïve Hearts and Proliferation Assay

Female C57BL/6 mice (6–8 weeks old) were purchased from Charles River (Sultzfield, Germany). Mice were anesthetized with 2%–3% isoflurane (ApoEx, Stockholm, Sweden), and hearts were extracted and rinsed in ice-cold Hank’s balanced salt solution (HBSS) to remove residual blood. The study was approved by the local Animal Ethics Committee in Gothenburg, Sweden (133-2013-08-21, 118-2014-06-25) and conducted in adherence with institutional animal use and Swedish national regulations on animal laboratory animal care. The hearts were minced into small pieces and digested into single cells with collagenase IV (1 mg/mL, 17104-019, Gibco, Life Technologies), dispase (0.15 mg/mL, 17105-041, Gibco, Life Technologies), and DNase (200 U/mL, D5025, Sigma Aldrich) for 45 min at 37 °C, then filtered through a 40-μm cell strainer (BD Falcon). The cell suspension was washed and then centrifuged at 300× *g* for 7 min to remove cell debris. The supernatant was aspirated, and the cell pellet was resuspended in ice-cold isolation buffer (PBS + 5% BSA + 2 mM EDTA). Sca1^+^/CD31^−^ cardiac progenitor cells were isolated with MACS (magnetic-activated cell sorting) cardiac progenitor cell isolation kit, Sca1 (130-098-374, Miltenyi Biotech Bergisch Gladbach, Germany), according to the manufacturer’s instructions. In brief, the isolation was performed in a two-step procedure. Firstly, the CD31^+^ cells were indirectly magnetically labeled with CD31–biotin, followed by anti-biotin monoclonal antibodies conjugated to micro beads. The labeled cells were subsequently depleted by separation in an LS column placed in the magnetic field of a MACS separator. In the second step, the Sca1^+^ cells were labeled with anti-Sca1 micro beads and isolated by two rounds of positive selection from the pre-enriched CD31^–^ cell fraction by separation over an MS column placed in the magnetic field of a MACS separator. Sca1^+^/CD31^−^ cells were counted and plated in 24-well plates (Corning) pre-coated with Cell Start (1:50 dilution, A10114201, ThermoFisher) in F12/DMEM culture medium (31331-028, ThermoFisher) supplemented with 10% FBS, 1% penicillin/streptomycin (PEST 15140-122, ThermoFisher), LIF (10 ng/mL, PMC9484, ThermoFisher), FGF2 (10 ng/mL), EPO (Erythropoietin; 0.005 U/mL, PHC2054), and beta-mercaptoethanol (0.1 mM, 31350-010). Cells were cultured at 37 °C for at least three days to allow cells to attach prior to medium refreshment and were passaged at 80%–90% confluence. To enrich for cardiogenic cardiac progenitor cells [49], PDGFRα-positive cells were isolated from the Sca1^+^/CD31^−^ population after at least one week in culture with MACS according to the manufacturer’s instructions (PDGFRα microbead isolation kit, 130-101-502, Miltenyi Biotech). Sca1^+^/CD31^−^/PDGFRα^+^ cells were used in proliferation experiments between passage 5 and passage 7. Cells were seeded at a density of 750 cells/well in 50 µL of culture medium in 384-well Corning plates coated with Cell Start (1:50 dilution). After 24 h, medium was removed and replaced with assay medium (F12/DMEM culture medium supplemented with 5% FBS and 1% PEST) containing the secretome proteins to be tested (2 µL). FGF2 (100 ng/mL) was used as a positive control and PBS was used as a neutral control. After 72 h, the cells were fixed in 4% formaldehyde for 20 min at RT, washed with PBS, and stained for nuclei with Hoechst 33342 (1:5000) for 10 min at RT, followed by final washing with PBS. The plates were imaged on an ImageExpress Micro (Molecular Devices) using 10× objective, and images were analyzed for nuclear counts using the MetaXpress software (Molecular Devices).

### 4.7. Human iPSC-Derived Cardiomyocyte Culture and Proliferation Screen

Cryopreserved human iPSC-derived human cardiomyocytes (CMC-100-012-000.5) were purchased from Cellular Dynamics International (CDI, Madison, WI, USA) and stored at −150 °C until use. Briefly, the cryopreserved iPSC cardiomyocytes were thawed and plated according to the manufacturer’s instructions. Cells were seeded in CellBind 384-well Corning plates (#3770) at a density of 4000 cells/well in 50 µL of seeding medium (Cellular Dynamics #M1001). The medium was changed to maintenance medium (Cellular Dynamics, #M1003) after 16–24 h. The cells were incubated for six days at 37 °C with medium change every second–third day. On day 6, the cells were dosed with proteins in supplemented (supplement B) Williams E medium together with EdU (2 µM). FGF2 (rh FGF2 PeproTech, # 100-18B) was used as a positive control. After 48 h of treatment, cells were fixed for 15 min at RT by adding 6 µL of 37% formaldehyde to each well without removing any medium and washed with PBS. EdU detection was done using the Click-it Alexa Fluor 488 kit (Thermo Fisher, #C10351). The plates were sealed and imaged on an ImageExpress Micro XLS (Molecular Devices) using 4× objective. Images were analyzed for EdU-positive nuclei compared to total nuclei using the MetaXpress software (Molecular Devices).

### 4.8. QCM Experiments

For the QCM interaction analysis, cells were grown on cell-optimized polystyrene 1 (COP-1) surfaces (Attana AB, Sweden, 3621-3033) before evaluation in an Attana Cell™ 200 biosensor. hCPCs were handled in hCPC assay medium, and hCFs were handled in William’s E medium. To coat the surface, 40,000 cells in 0.7 mL of medium were added to the cell chamber, which was incubated at 37 °C (5% CO_2_ and 95% humidity) for 18 h before cells were rinsed three times with 0.7 mL of PBS at RT. Cells were stabilized in fresh 4% methanol-free formaldehyde for 15 min at 4 °C, and cell coverage was determined by staining and visualization of cells under a fluorescent microscope with 3 µM DAPI (4′,6-diamidino-2-phenylindole; D9542, Merck). Sensor surfaces were inserted in the biosensor or stored under humidity at 4 °C in the dark until use.

For the interaction between proteins and cell surfaces, all experiments were performed under continuous flow, flow rate 20 µL/min at 37 °C, with an injection time of 105 s for each protein. A blank injection of running buffer, performed before each protein injection, was subtracted from the subsequent protein injection to correct for baseline drift. Also, surfaces were left to reach the baseline after protein injections where the signal was low due to little protein binding, whereas surfaces were regenerated when there was strong protein binding using a 30-s injection of 10 mM glycine (pH 2). Generally, four concentrations were used for each protein, and each concentration was injected in duplicates. To assess specificity of binding, experiments were performed at a flow rate of 10 µL/min at 37 °C. In all cases, experiments were performed on two independent cell-covered COP-1 surfaces. Saturation levels for FGF9 and FGF16 were estimated based on equilibrium studies of high protein concentration using sequential injection. In brief, FGF9 (or FGF16) was injected until equilibrium was established over cells grown on COP-1 surface. Subsequently, a mixture of FGF9 and FGF16 was injected over the equilibrated surface. The frequency change in the sensor surface resonance (ΔF) during the binding experiments was recorded using the Attester software (Attana AB), and the data were analyzed using the Evaluation (Attana AB) and TraceDrawer software (Ridgeview Instruments AB, Stockholm, Sweden) using 1:1 or 1:2 binding models to calculate the kinetic parameters, including the rate constants (K_a_, K_d_), dissociation equilibrium constant (K_D_), and the maximum binding capacity (B_max_).

### 4.9. qPCR Experiments on hCFs and hCPCs

RNA was isolated from iPS-derived hCPCs (two donors) and hCFs (three donors) using the RNeasy Mini kit (QIAGEN) and subsequently reverse-transcribed into complementary DNA (cDNA) with a High-Capacity cDNA Reverse Transcription Kit (Applied Biosystems) according to the manufacturer’s instructions. For qRT-PCR, the samples were analyzed using the Quantstudio 7 Flex (Applied Biosystems) in triplicate, and the following TaqMan assays were used: Hs00241111_m1 (*FGFR1*), Hs01552926_m1 (*FGFR2*), Hs00179829_m1 (*FGFR3*), and Hs01106908_m1 (*FGFR4*). The expression levels of the FGF receptors were compared to the endogenous control 60S acidic ribosomal protein P0 (*RPLP0*), and evaluated by the comparative change in threshold cycles, using the 2^−ΔCt^ formula.

## 5. Conclusions

From the unbiased screen of 923 human secretome recombinant proteins on human hCPCs, we identified FGF16 as an interesting candidate for cardiac regeneration and repair. FGF16 specifically induced proliferation of human CPCs and cardiomyocytes but not human fibroblasts. Evidence from the literature also shows that FGF16 has an important role in regulating the cardiomyocyte number during heart development. Altogether, FGF16 is an interesting target to explore further.

## Figures and Tables

**Figure 1 ijms-20-06037-f001:**
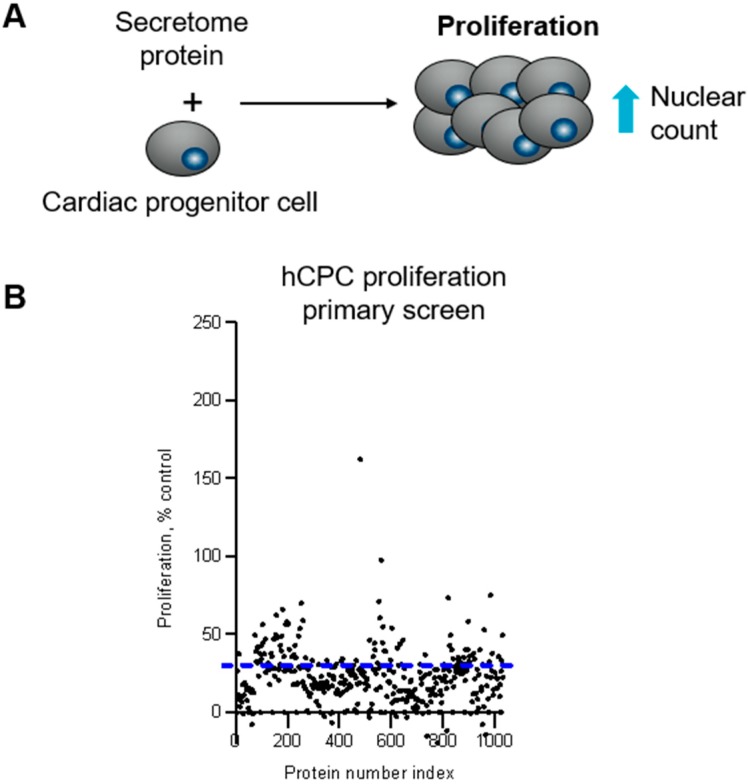
Proliferation of cardiac progenitor cells (**A**) The secretome library was screened to identify proteins that stimulate cell proliferation of hCPCs (**B**) The secretome library (923 unique proteins) was screened in three-point CR in duplicate using iPS-derived hCPCs. The effect of secretome proteins on hCPC proliferation was measured as a change in cell number determined by nuclei count and normalized as percentage activity based on the on-plate PBS neutral controls. The blue line indicates the cut-off for actives in the primary screen. The data shown are from one concentration screened.

**Figure 2 ijms-20-06037-f002:**
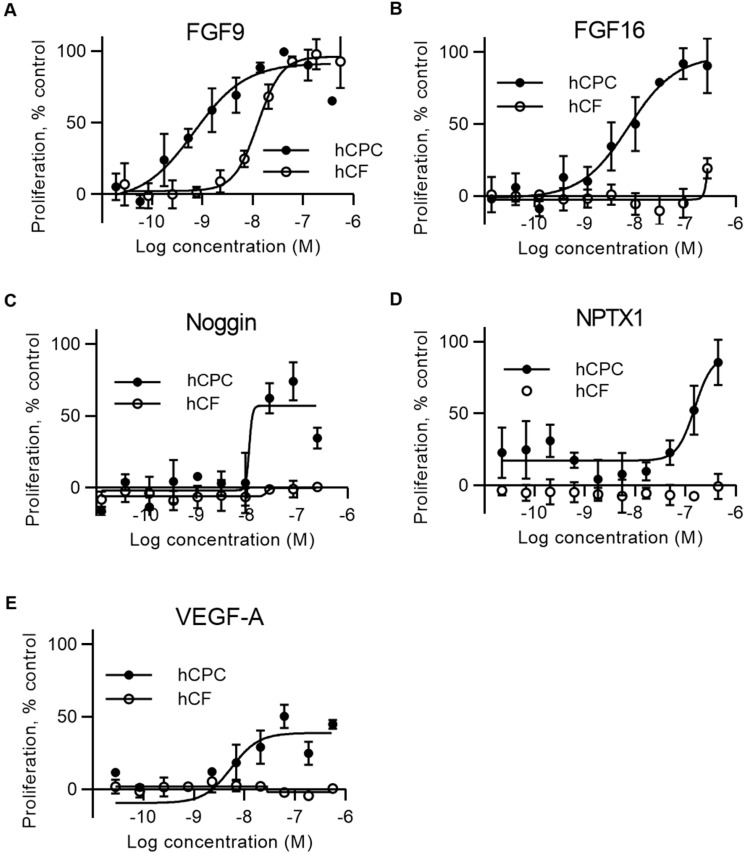
FGF16, Noggin, VEGF-A and NPTX1 specifically induce proliferation of hCPCs, whereas FGF9 also induces proliferation of hCFs. Concentration–response curves showing proliferative activity of identified proteins on hCPCs and hCFs (**A**–**E**). Proliferative activity of FGF9 (**A**), FGF16 (**B**), Noggin (**C**), NPTX1 (**D**), and VEGF-A (**E**) on hCPCs and hCFs. The effect of proteins on hCPC (black circles) and hCF (white circles) was measured as a change in cell number determined by nucleus count. Data are means ± SD from three independent samples. Statistical significance information is presented in Appendix A.

**Figure 3 ijms-20-06037-f003:**
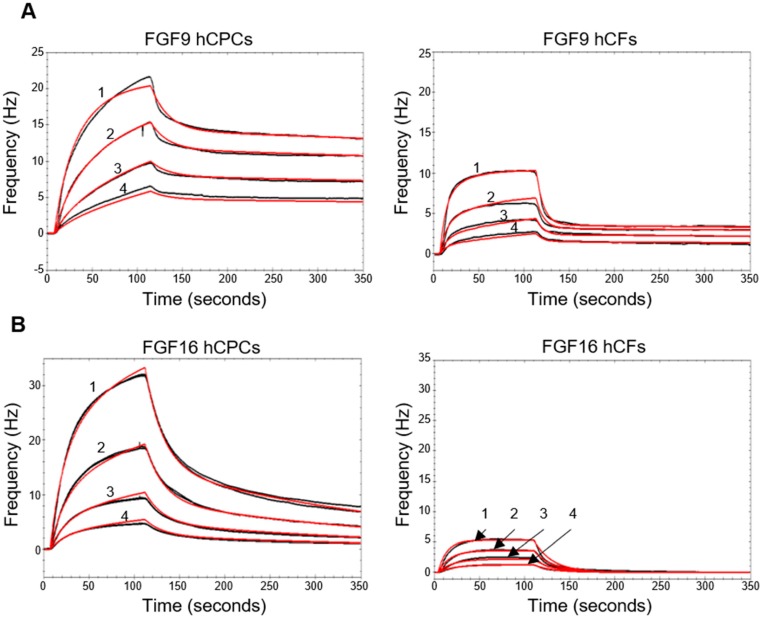
Biophysical characterization of the interaction of FGF16 and FGF9 with hCPCs and hCFs. Kinetic evaluation of the interactions between FGF9 (**A**) or FGF16 (**B**) and hCPCs or hCFs using QCM technology. (**A**) Human FGF9 was injected over the hCPC (left trace) or hCF (right trace) surface, and the responses were recorded (black lines). Numbering 1–4 corresponds to 34, 17, 8, and 4 µM protein, respectively. (**B**) Human FGF16 was injected over the hCPC (left) or hCF (right) surface, and the responses were recorded (black lines). Numbering 1–4 corresponds to 1, 0.5, 0.25, and 0.125 µM protein, respectively. Theoretical fits were overlaid (red lines). Chi^2^ for the global fit was, from top to bottom, 0.12, 0.04, 0.14, and 0.02, respectively. The association and dissociation rate constants, as well as the affinity constant, were obtained (summarized in Appendix A).

**Figure 4 ijms-20-06037-f004:**
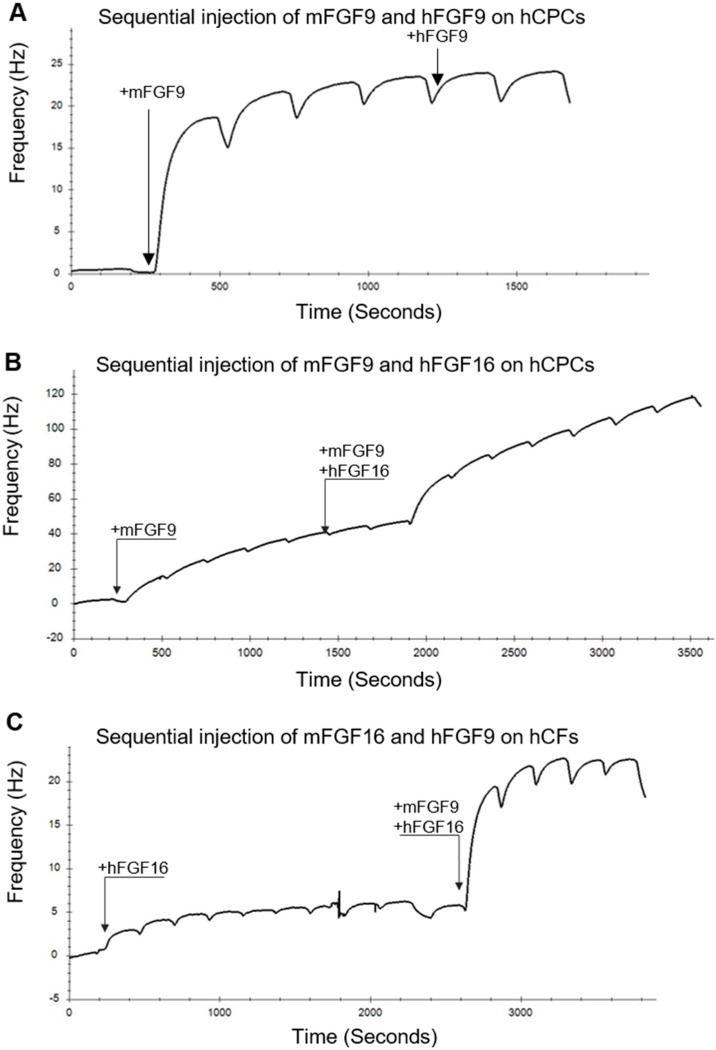
Biophysical characterization of interaction of FGF16 and FGF9 with hCPCs and hCFs suggests that the proteins bind to different receptors. (**A**) Kinetic evaluation of the sequential interactions between mouse and human FGF9 and hCPCs. mFGF9 was repeatedly injected over the hCPCs. At ~1200 s, hFGF9 was injected. No additional response was detected. (**B**) mFGF9 was repeatedly injected over the hCPCs. At ~1900 s, a mixture of mFGF9 and hFGF16 was injected. (**C**) hFGF16 was repeatedly injected over the hCFs. At ~2600 s, a mixture of mFGF9 and hFGF16 was injected. In (**B**) and (**C**), the additional response from the mixture was similar to the response elicited by the first ligand alone. FGF9 and FGF16 were used at 34 and 1 µM, respectively.

**Figure 5 ijms-20-06037-f005:**
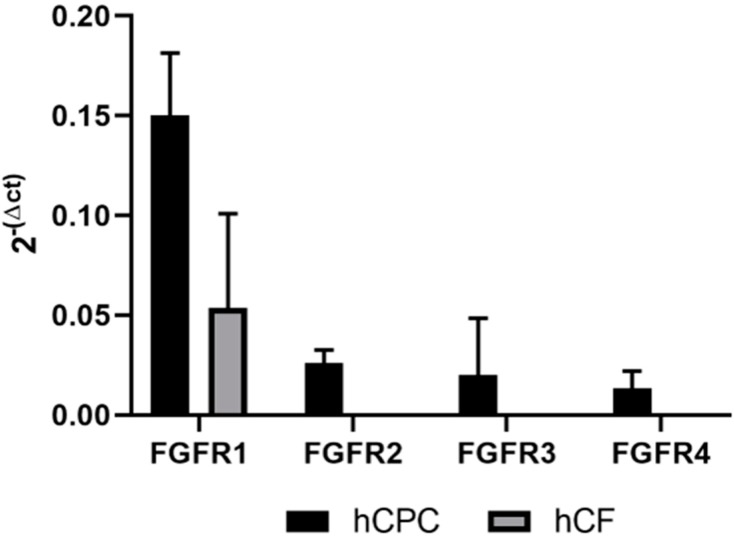
*FGFR1* is most abundantly expressed on hCPC and hCFs, followed by *FGFR2* specifically on hCPCs. Messenger RNA (mRNA) expression levels of FGF receptors in iPS-derived hCPCs and hCFs, detected by qRT-PCR analysis. *FGFR1* shows detectable mRNA levels in both cell types, while *FGFR2*, 3, and 4 only show specific expression in iPS-derived hCPCs. Each receptor expression is compared to the endogenous control 60S acidic ribosomal protein P0 (*RPLP0*) and reported as 2^−ΔCt^ values. Data are means ± SD from three individual experiments.

**Figure 6 ijms-20-06037-f006:**
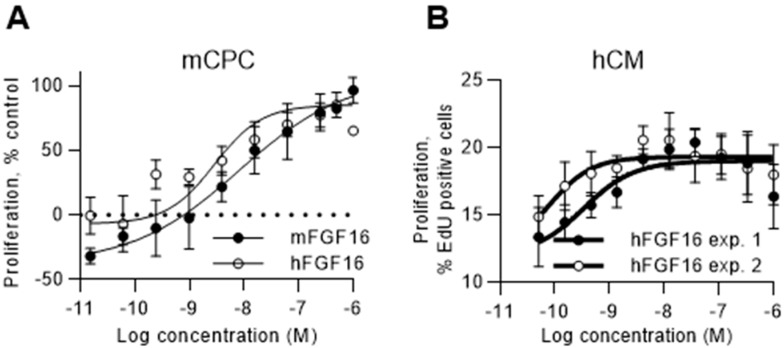
FGF16 induces proliferation of naïve mouse CPCs and human cardiomyocytes (**A**) Mouse and human FGF16 induces proliferation of mouse naïve CPCs (Sca1^+^, PDGFRa^+^, CD31^−^ (cluster of differentiation 31)) isolated from mouse hearts, as described in Section 4. Proliferation was measured as an increase in nuclear count, as described in Section 4. (**B**) Human FGF16 induces proliferation of iPS-derived human cardiomyocytes. Proliferation was measured as an increase in EdU incorporation, as described in Section 4. Data are means ± SD from three independent samples for mCPC, and six independent samples for hCM. Statistical significance information is presented in Appendix A.

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
