# Peer review of "Phenotypic Screen with the Human Secretome Identifies FGF16 as Inducing Proliferation of iPSC-Derived Cardiac Progenitor Cells"

_ijms, 2019, doi:10.3390/ijms20236037_

Round 1
Reviewer 1 Report
This report by Jennbacken et al. describes the discovery of FGF16 as a chemokine inducer of cardiac progenitor cell (CPC)-specific proliferation. The authors performed a medium-throughput assay to screen 923 secretome proteins against human iPSC-derived CPCs and human cardiac fibroblasts (hCFs). FGF16 was more active in inducing the hCPCs versus the hCFs. Kinetic studies supported FGF16 binding to hCPCs, possibly through FGF1 (which was identified through qPCR). The CPC proliferation potential of FGF16 was verified in CPCs isolated from mouse hearts. Additionally, cardiomyocyte proliferation was induced by FGF16.
Overall, this was a good screening study to find novel agents that promote cardiac progenitor cell proliferation toward heart repair and regeneration, and especially to affect cardiomyocyte replacement, since induced paracrine factors tend to promote endothelial and vascular cell proliferation. This study supports that FGF16 is a promising candidate for further study for its activity and CPC and cardiomyocyte specificity.
In my opinion, a major deficiency lies in not performing even a preliminary assessment of potential differentiation phenotypes of the CPCs treated with FGF16. Since FGF16 is a fibroblast growth factor, do the CPCs begin to show fibroblast-like markers? CPCs are, by definition, progenitor cells, and do not have to become cardiomyocytes; they may differentiate into fibroblasts and other cardiac cells. Therefore, since the cells were analyzed microscopically for proliferation, I believe an assessment of differentiation morphology should have been considered. At the very least, the authors should address CPC-plus-FGF16 differentiation potential in the Discussion section. I acknowledge the brief mention that Nkx2.5 immunofluorescence staining was used to verify that the CPCs “had not differentiated into any other cell type,” in the Materials and Methods section, but this was not sufficiently fleshed-out elsewhere.
In terms of minor issues, please note the following:
“Datum” is the singular form and “data” is plural. Therefore, phrases like “the data is presented” is grammatically incorrect and should be “the data are presented.” This grammatical mistake is found throughout the manuscript and should be corrected. “Medium” is the singular form and “media” is plural. Therefore, phrases like “the media was removed” is grammatically incorrect and should be “the medium was removed,” when you are discussing only one medium type (e.g. hCPC assay medium). This grammatical mistake is found throughout the manuscript, especially in the Materials and Methods section, and should be corrected. In Fig. 1B, the axis label “Index” is not self-explanatory and should be changed to something like “Protein Identification Number.” Also, adding the threshold line, showing where the cut-off was for compounds that were considered active, would be helpful. Were statistics performed on any of your data, especially those presented in Fig. 5? Line 246: correct “fibrosis was decreased i” to “fibrosis was decreased in.” Materials and Methods: please explain why the protein concentrations in the assay couldn’t be normalized and equal. You state using an “average top-concentration” of 200-400 nM. Two-fold differences in starting concentrations seem problematic to me. Materials and Methods: You used Nkx2.5 immunofluorescence staining to determine the cells hadn’t differentiated into any other cell type. However, couldn’t they have become cardiomyocyte-like and still express Nkx2.5? What cell type would lose Nkx2.5 expression that these cells could plausibly become?Author Response
First reviewer comments:
In my opinion, a major deficiency lies in not performing even a preliminary assessment of potential differentiation phenotypes of the CPCs treated with FGF16. Since FGF16 is a fibroblast growth factor, do the CPCs begin to show fibroblast-like markers? CPCs are, by definition, progenitor cells, and do not have to become cardiomyocytes; they may differentiate into fibroblasts and other cardiac cells. Therefore, since the cells were analyzed microscopically for proliferation, I believe an assessment of differentiation morphology should have been considered. At the very least, the authors should address CPC-plus-FGF16 differentiation potential in the Discussion section. I acknowledge the brief mention that Nkx2.5 immunofluorescence staining was used to verify that the CPCs “had not differentiated into any other cell type,” in the Materials and Methods section, but this was not sufficiently fleshed-out elsewhere.
Response:
Thanks for the comment. We agree that it would have been interesting to examine the differentiation potential of human CPCs in response to FGF16, but we believe that the short time of incubation after addition of secretome proteins, 3 days, is not sufficient to differentiate the cells into other cell types.
In this work we tested the effect of FGF16 on CPC proliferation in an assay that was optimized to run over 3 days. At plating and also three days post plating a majority of the cells stain positive for Nkx2.5 which indicate that they still are in the cardiac progenitor cell stage. In previous work we developed a high content screening assay for identification of compounds that can induce CPC differentiation. CPCs can be stimulated to differentiate into cardiomyocytes, endothelial cells and smooth muscle cells but would need at least 7 days in culture to detect markers for these cell types [1]. The optimized assay conditions for CPC differentiation into cardiomyocytes include addition of XAV939 (which is a strong inducer for cardiomyocyte differentiation). cTnT expression was first detected at day 4 after plating and the fraction of cTnT positive cells increased up to day 7 of differentiation. 10% of the cTnTneg population expressed smooth muscle actin (SMA). In addition, when cultured with FGF2 or VEGF-A, CD31 could be detected on a large proportion of the cells, indicating endothelial cell development [1]. As the reviewer points out, CPCs also have the capability to differentiate into fibroblasts which was demonstrated in a recent paper by Zhang et al, [2]. Thus, depending on the differentiation stimuli, CPCs can give rise to various cell types. Similar to our work, Zhang et al show that the time in culture needed is long, in this study 18 days. Based on the above, we didn't check whether FGF16 induced differentiation to other cell types in the experiments presented in the submitted manuscript. However, since we only check for Nkx2.5 and no other cell type specific markers, we cannot exclude the possibility that FGF16 could induce differentiation of CPCs to fibroblasts or other cell types during the assay period. We agree with the reviewer that it is of importance to discuss this in more detail, and we have added a section to the discussion (lines 220):
We show that FGF16 specifically induced proliferation of CPCs and induced cell cycle activity in human cardiomyocytes. The data highlight a novel effect on CPCs and show that FGF16 overall is an interesting factor to explore in cardiac regeneration and repair, since it also stimulates cardiomyocytes. In contrast, FGF9 had a more general proliferative effect on cardiac progenitor cells, cardiac fibroblasts and kidney fibroblasts (Figure 2 and unpublished results). We have previously used human iPS derived CPCs in phenotypic high-content screening assays in our team to identify small molecule compounds that induce proliferation [1] [3]. In the presented work, we focused on screening a human secretome library in a proliferation assay with hCPCs. It is unlikely that the hCPC differentiated into other cell types during this time frame, and Nkx2.5 stainings were performed on selected proteins, also confirming that a majority of cells at 3 days post plating were at the cardiac progenitor cell stage. However, we cannot exclude the possibility that secretome proteins induced differentiation into other cell types. CPC differentiation into cardiomyocytes, smooth muscle cells, endothelial cells or fibroblasts can be captured, but requires an extended incubation for at least 7 days [1] [2].
In terms of minor issues, please note the following:
“Datum” is the singular form and “data” is plural. Therefore, phrases like “the data is presented” is grammatically incorrect and should be “the data are presented.” This grammatical mistake is found throughout the manuscript and should be corrected. “Medium” is the singular form and “media” is plural. Therefore, phrases like “the media was removed” is grammatically incorrect and should be “the medium was removed,” when you are discussing only one medium type (e.g. hCPC assay medium). This grammatical mistake is found throughout the manuscript, especially in the Materials and Methods section, and should be corrected. In Fig. 1B, the axis label “Index” is not self-explanatory and should be changed to something like “Protein Identification Number.” Also, adding the threshold line, showing where the cut-off was for compounds that were considered active, would be helpful. Were statistics performed on any of your data, especially those presented in Fig. 5? Line 246: correct “fibrosis was decreased i” to “fibrosis was decreased in.” Materials and Methods: please explain why the protein concentrations in the assay couldn’t be normalized and equal. You state using an “average top-concentration” of 200-400 nM. Two-fold differences in starting concentrations seem problematic to me. Materials and Methods: You used Nkx2.5 immunofluorescence staining to determine the cells hadn’t differentiated into any other cell type. However, couldn’t they have become cardiomyocyte-like and still express Nkx2.5? What cell type would lose Nkx2.5 expression that these cells could plausibly become?
Response:
We have reviewed the whole manuscript, and Data are now used in plural form in the manuscript, Medium is used for descriptions of single use 1B has been changed as suggested by the reviewer: the axis label is changed to “Protein number index” and a threshold line has been added. The legend to Figure 1B is also modified to reflect the change. Statistics have been included for Figure 5. The legend to Figure 5 is also modified to reflect the change. Line 246: correct “fibrosis was decreased i” to “fibrosis was decreased in”. This has been changed (line 264). ” Materials and Methods: please explain why the protein concentrations in the assay couldn’t be normalized and equal. You state using an “average top-concentration” of 200-400 nM. Two-fold differences in starting concentrations seem problematic to me.Response:
We apologize that this section is slightly confusing and there is also an error in the text. The produced proteins vary in starting concentration due to some proteins being produced in lower quantities. All proteins that were included in the screen had a concentration >2 µM. However, 90% of proteins had a starting concentration of 9 µM. (After production, all proteins that had a concentration > 9 µM were diluted to 9 µM). As a consequence, the top concentration in the primary screen was 450 nM for a majority of proteins. For a small fraction of proteins, the starting concentration varied between 80 nM-450 nM. We cannot exclude that there are false negatives in the screen due to low concentration of protein. However, this is not different to other described secretome screens [4] [5]. The experimental setup presented is in our view unique since we are using purified proteins, at known concentrations and with a high starting concentration in the primary screen. The text has been changed to correct the error (lines 322-323):
However, all proteins with a concentration above 9 µM were diluted to 9 µM. Based on these numbers the average top-concentration of protein in the cell assays was ~400 nM.
Materials and Methods: You used Nkx2.5 immunofluorescence staining to determine the cells hadn’t differentiated into any other cell type. However, couldn’t they have become cardiomyocyte-like and still express Nkx2.5? What cell type would lose Nkx2.5 expression that these cells could plausibly become?
Response:
Yes, we agree with your comment. Nkx2.5 could still be expressed on the cell even if it had differentiated into a cardiomyocyte and we know that that is the case based on previous data that was published in Stem Cells Translational Medicine in 2016 by Drowley et al. [1]. However, 3 days after plating the CPCs are still at a progenitor cell stage since TnT starts to be expressed at the earliest 4 days post plating (based on data from the above publication). We have addressed this comment in more detail above. We have also added a section to the discussion to clarify.
Drowley L, et al., Human induced pluripotent stem cell-derived cardiac progenitor cells in phenotypic screening: A transforming growth factor-beta type 1 receptor kinase inhibitor induces efficient cardiac differentiation. Stem Cells Transl Med, 2016. 5(2): p. 164-174. Zhang H, et al., Generation of quiescent cardiac fibroblasts from human induced pluripotent stem cells for in vitro modeling of cardiac fibrosis. Circ Res, 2019. 125(5): p. 552-566. Drowley L, et al., Discovery of retinoic acid receptor agonists as proliferators of cardiac progenitor cells through a phenotypic screening approach. Stem Cells Transl Med, 2019. Lin H, et al., Discovery of a cytokine and its receptor by functional screening of the extracellular proteome. Science, 2008. 320(5877): p. 807-811. Gonzalez R, et al., Screening the mammalian extracellular proteome for regulators of embryonic human stem cell pluripotency. Proc Natl Acad Sci U S A, 2010. 107(8): p. 3552-3557.
Reviewer 2 Report
Jennbacken et al. introduce a screen of a human secretome library in a phenotypic screen using human cardiac progenitor cells focussing on cell proliferation measured by nuclear count. Thereby, they show that FGF1, FGF4, FGF9, FGF16, FGF18 and a number of additional proteins induce proliferation of the CPCs. Of note, FGF16 is specific for proliferation of CPCs while FGF9 also led to proliferation of human cardiac fibroblasts. Moreover, biosensor analysis of receptor preferences and quantification of receptor abundances suggested that FGF16 and FGF9 bind to different FGF receptors on the cardiac progenitor cells and cardiac fibroblasts. In addition, FGF16 also proliferated native cardiac progenitor cells isolated from mouse heart and human cardiomyocytes obtained from iPSCs. The authors conclude that FGF16 could be a used as a tool to induce cardiac regeneration and repair.
I find the paper interesting and of relevance for the field. The presented experiments appear to be well performed. Yet, I still have the following point which should be addressed:
1) Please indicate statistical significance in Figs. 2, 6, 5 and S1.
2) Terminally, application of FGF16 in patients would obviously be the ultimate goal. In this regard a preclinical model (e.g. infarcted mouse or rat hearts) would be desirable. I accept that this would, however, exceed the present study. Yet at least the authors should investigate the integrity oft he cardiomyocytes obtained after boosted proliferation vie the growth factor. This could ideally be done via physiological analyses (e.g. Patch-Clamping, MEA-analyses in combination with some standard pharmacology). In addition some immune-stainings adressing sarcomere-integrity should be added.
Round 2
Reviewer 1 Report
I believed that the authors have addressed the reviewer concerns satisfactorily. I now recommend accepting the manuscript with no further changes.
Reviewer 2 Report
Thank you very much. You have adressed my points.